# Research Updates of Plasmid-Mediated Aminoglycoside Resistance 16S rRNA Methyltransferase

**DOI:** 10.3390/antibiotics11070906

**Published:** 2022-07-07

**Authors:** Weiwei Yang, Fupin Hu

**Affiliations:** 1Institute of Antibiotics, Huashan Hospital, Fudan University, Shanghai 200040, China; 21211220038@m.fudan.edu.cn; 2Key Laboratory of Clinical Pharmacology of Antibiotics, Ministry of Health, Shanghai 200040, China

**Keywords:** aminoglycosides, antimicrobial resistance, 16S rRNA methyltransferases

## Abstract

With the wide spread of multidrug-resistant bacteria, a variety of aminoglycosides have been used in clinical practice as one of the effective options for antimicrobial combinations. However, in recent years, the emergence of high-level resistance against pan-aminoglycosides has worsened the status of antimicrobial resistance, so the production of 16S rRNA methyltransferase (16S-RMTase) should not be ignored as one of the most important resistance mechanisms. What is more, on account of transferable plasmids, the horizontal transfer of resistance genes between pathogens becomes easier and more widespread, which brings challenges to the treatment of infectious diseases and infection control of drug-resistant bacteria. In this review, we will make a presentation on the prevalence and genetic environment of 16S-RMTase encoding genes that lead to high-level resistance to aminoglycosides.

## 1. Introduction

Aminoglycoside antibiotics were discovered and isolated from soil *Actinobacteria* in the 1940s, and streptomycin was the first aminoglycoside antibiotic used in the clinical treatment of tuberculosis and Gram-negative bacteria infection [1]. The mechanism of aminoglycosides is to bind the A site of 16S rRNA, which consists of the 30S ribosomal subunit, leading to the inhibition of protein synthesis and bacteria death [2]. Therefore, aminoglycosides have a broad spectrum of antimicrobial activity against both Gram-positive and Gram-negative bacteria and are often used in combination with β-lactam antibiotics, especially the third-generation cephalosporins [3].

Since the 1980s, due to the side effects of ototoxicity and nephrotoxicity, aminoglycosides have been used less and less in clinics and gradually replaced by β-lactams and quinolones, which are less toxic and have a wider antimicrobial spectrum [4]. However, with the increase of β-lactams and quinolone antimicrobial resistance and the prevalence of multidrug-resistant bacteria, the retained potency of aminoglycoside antibiotics has renewed interest in their use in clinical practice [5]. Additionally, aminoglycosides can be used as one of the effective antimicrobial combinations for the treatment of life-threatening infections caused by multidrug-resistant bacteria. Because aminoglycosides can be used in combination with other antibiotics, the toxicity can be decreased by adjusting the dosage [6]. However, in recent years, reports about high-level resistance to aminoglycosides have increased and spread widely around the world. In this review, we will make a presentation on the prevalence and genetic environment of 16S-RMTases, which lead to high-level resistance to pan-aminoglycosides.

## 2. Antimicrobial Resistance Mechanism of Aminoglycosides

According to the difference in chemical structures, aminoglycosides can be divided into 4,5-disubstituted 2-deoxystreptamine (DOS), such as neomycin and paromomycin; 4,6-disubstituted 2-DOS, such as gentamicin, amikacin, kanamycin, arbekacin, and tobramycin; monosubstituted DOS, such as apramycin; and no DOS ring, such as streptomycin [1,4]. The mechanisms of aminoglycosides resistance include: (1) modification or inactivation of aminoglycosides modifying enzymes; (2) increased expression of efflux pump; (3) decreased drug permeability; and (4) modification of the drug target, such as generating 16S-RMTases that interfere with binding of the aminoglycosides [7]. Numerous studies have shown that aminoglycosides modifying enzymes and 16S-RMTases can lead to high-level resistance to aminoglycosides in bacteria. Aminoglycosides modifying enzymes are the most common mechanism of aminoglycosides resistance in bacteria, consisting of aminoglycoside acetyltransferases, aminoglycoside phosphotransferases, and aminoglycoside nucleotidyltransferases [5]. The above three aminoglycoside modification enzymes can catalyze the modification of different OH and NH_2_ groups of 2-DOS molecules. Because the encoding genes of aminoglycoside modifying enzymes often occur with mutations, many subclasses of the enzymes are produced, making more aminoglycosides become the substrates of the modifying enzymes and increasing the antimicrobial resistance [8].

Although aminoglycoside modification enzymes are the most common resistance mechanism, due to the specificity of substrates, they cannot mediate high-level resistance to multiple aminoglycosides [9]. However, another mechanism of bacterial resistance to aminoglycosides, modification of drug target, leads to high-level resistance against pan-aminoglycoside, with minimal inhibitory concentration (MIC) > 256 μg/mL. There are two ways to modify the drug target: post-transcriptional methylation of 16S rRNA to block aminoglycosides binding to the target [7]; gene point mutation (nucleotide substitution), for example, *Mycobacterium tuberculosis rpsL* gene encodes S12 protein to mediate streptomycin resistance [10].

## 3. Plasmid-Mediated 16S rRNA Methylase Resistance Gene and Its Transfer Mechanism

Acquired 16S rRNA methylase is the most clinically significant aminoglycoside resistance mechanism. The concrete resistance mechanism is that with the catalysis of 16S-RMTase, adding a CH_3_ group provided by S-adenosine methionine (SAM) to specific residues at the A site of 16S rRNA, the binding ability of methylated 16S rRNA to aminoglycosides is significantly reduced, resulting in extensive and high-level resistance to various aminoglycosides [4]. The post-transcriptional modification of 16S rRNA is intrinsic to Actinomycetes (e.g., *Streptomyces* and *Micromonospora*) species that produce aminoglycosides to protect themselves from the damage of endogenous aminoglycosides [11,12].

Since the first acquired 16S-RMTase gene was identified in 2003, new genotypes and subtypes have emerged continuously, mediating high-level resistance to a variety of aminoglycosides. So far, a total of eleven acquired 16S-RMTases have been identified and reported, which include ArmA, NpmA, NpmB, and RmtA through RmtH [4,12]. In order to figure out the evolutionary relationships of 16S-RMTases encoding genes and their subtypes, a phylogenetic tree is reconstructed and shown in Figure 1. About the enzymatic function of 16S-RMTases, different types of methyltransferases have different action sites of 16S rRNA: ArmA, RmtA through RmtH methylate N7-G1405 of 16S rRNA and confer high-level resistance to 4,6-disubstituted 2-DOS; NpmA and NpmB methylate N1-A1408 of 16S rRNA and are resistant to 4,5-disubstituted 2-DOS, 4,6-disubstituted 2-DOS, and monosubstituted DOS but only susceptible to streptomycin [1,12,13]. Although the action site is kind of different, the whole types of methyltransferases are able to mediate high-level resistance to pan-aminoglycosides.

Except for the high-level antimicrobial resistance, the 16S-RMTase encoding genes found so far are mostly located within transferable plasmid and/or associated with mobile genetic elements such as transposons, integrons, and insertion sequence [14]. Furthermore, with the support of plasmid and mobile genetic elements, 16S-RMTase encoding genes are often associated with other resistance genes, leading multi-drug resistance even pan-drug resistance. Because of the high-level antimicrobial resistance and the mobility conferred by plasmids as well as other mobile genetic elements, we need pay more attention to the pan-aminoglycoside resistance caused by 16S-RMTase. Here are detailed descriptions of each 16S-RMTase encoding gene and its transfer mechanism; epidemic strains carrying 16S-RMTase encoding genes and their distribution are shown in Table 1, and the genetic context of the 16S-RMTase encoding genes are shown in Figure 2.

### 3.1. ArmA

The 16S-RMTase gene *armA* was first identified on a transferable plasmid of *K. pneumoniae* in French [107]. Subsequently, *armA* has been widely spread throughout the world, primarily found in *A. baumanii*, *P. rettgeri*, and *K. pneumoniae*, especially prevalent in *A. baumanii*. What is more, *armA* often came along with the *rmtB* gene and led to high-level resistance to aminoglycosides with MIC ≥ 256 μg/mL [108,109,110,111]. Research studies showed that *armA* was often located on transferable plasmids belonging to different Inc types [19], and insertion sequences such as IS*26* might be involved in the mobilization of resistant genes around pathogens. In addition, these 16S-RMTases producers also showed resistance to β-lactams and carbapenems through various antimicrobial resistance genetic determinants, such as *bla*_TEM-1_, *bla*_CTX-M_, *bla*_NDM_, and other resistance genes [112,113]. The example of *armA* and its genetic context is shown in Figure 2.

### 3.2. RmtA

The 16S-RMTase gene *rmtA* was first identified in *P. aeruginosa* isolated from Japan in 2003 [9]. Then, *rmtA* gene was also detected among *P. aeruginosa*, *E. cloacae*, and *K. pneumoniae*, which were isolated from Korea, China, and Switzerland, mediating high-level resistance to aminoglycosides with MIC ≥ 512 μg/mL [9,29,31,114]. Further research studies showed that the type of plasmids carrying *rmtA* was usually IncA/C [29]. According to the genetic environment around the *rmtA*, it was speculated that *rmtA* was located on mercury-resistant transposons Tn*5041* and mobile genetic elements, such as κ-λ elements and IS*6100*, flank the *rmtA* gene and mediate horizontal transfer and homologous recombination between strains [114]. In Figure 2, you can acquire more details of the *rmtA* and its genetic context which is shown as an example. Several studies had also confirmed that the *rmtA* gene was usually associated with *bla*_CTX-M-15_, *bla*_NDM-1_, *bla*_SHV_, *bla*_TEM-1_, and other resistant genes, which largely limits the treatment of multidrug-resistant bacteria [9,114,115].

### 3.3. RmtB

The 16S-RMTase gene *rmtB* was first detected in *S. marcescens* isolated from French in 2004 [116]. *rmtB* is primarily found in *E. coli* and *K. pneumoniae* all over the world and results in resistance to aminoglycosides with MIC ≥ 512 μg/mL [7,51,54,117]. In China, *rmtB* was the main prevalent genotype of 16S-RMTase and was often combined with *armA*. Recently, with the development of molecular diagnostic techniques, several alleles of *rmtB* have been identified: *rmtB2* was found in *K. pneumoniae* and *P. rettgeri* [42,103]; *rmtB3* and *rmtB4* were found in *P. aeruginosa* [92]. Compared with the sequence of *rmtB1* (GenBank: NG_048058.1), *rmtB2* (GenBank: NG_048059.1) showed 96.6% nucleotide identity (26 nucleotides of difference), *rmtB3* (GenBank: NG_051535.1) showed 99.6% nucleotide identity (3 nucleotides of difference), and *rmtB4* (GenBank: NG_051536.1) showed 97.0% nucleotide identity (23 nucleotides of difference). Compared with RmtB1, the amino acid sequence of RmtB2 had 6 amino acid substitutions: Ala41Thr, Val124Ile, Val132Ile, Thr166Ile, Ile194Leu, and Thr229Ala; RmtB3 had 1 amino acid substitution: Ala82Val; RmtB4 had 4 amino acid substitutions: Val124Ile, Val132Ile, Thr166Ala, and Ile194Leu. Further studies of its genetic environment show that the *rmtB* gene is usually located on transposons of transferable plasmids with insertion sequences such as IS*CR1*, IS*Cfr1*, and IS*26*, leading to the transfer and transmission of *rmtB* much easier among different strains [118,119]. A typical example of *rmtB* and its genetic environment is displayed in Figure 2.

### 3.4. RmtC

The 16S-RMTase gene *rmtC* was firstly detected in *P. mirabilis* isolated from Japan in 2004 [63]. Since then, *rmtC* has been spread widely around the world and distributed in various species, such as *K. pneumoniae*, *E. coli*, *P. aeruginosa,* and *A. baumanii* [24,60,69], resulting in high-level resistance to aminoglycosides with MIC ≥ 1024 μg/mL [63]. According to the genetic sequences around the *rmtC*, there was an obvious pattern that IS*Ecp1* was located upstream of *rmtC* and played an important role in the expression of *rmtC* as well as the transfer among different Gram-negative strains [60,63]. This IS*Ecp1* element belonged to the IS*1380* family which was located at the ends of *rmtC* and contains a transposase gene (*tnpA*) and provided a promoter activity for expression of the adjacent *rmtC*. This structure enabled the *rmtC* gene to be transposed onto another plasmid [120,121]. As you can see, in Figure 2, we show an instance of *rmtC* and its genetic environment to help you understand the above statement. Recently, RmtC methyltransferase has been reported in China and isolated from *S. stanley* and *K. aerogenes* respectively. Further studies showed that the multidrug resistance regions and their genetic environment of the two strains’ plasmid were homologous to some extent, associated with carbapenemase and β-lactamase resistance genes such as *bla*_NDM-1_ and *bla*_CMY-6_ to mediate multidrug resistance [68,73].

### 3.5. RmtD

The 16S-RMTase gene *rmtD* was first identified in *P. aeruginosa* isolated from Brazil in 2007, mainly distributed in South America (such as Brazil, Chile, Argentina, etc.), commonly found in *P. aeruginosa* and *K. pneumoniae* [77,78,79]. Consistent with the characteristics of 16S-RMTase, the strains carrying *rmtD* show a high level of resistance to aminoglycosides with MIC ≥ 256 μg/mL, even *rmtD3* mediates MIC of aminoglycosides up to 1024 μg/mL [75,81].

What is more, several alleles of *rmtD* have continually been reported in recent years. *rmtD2* was isolated from the plasmid of *E. aerogenes* and *C. freundii* in Argentina [79] and displayed 97.3% nucleotide identity (20 nucleotides of difference) and 96.4% amino acid identity (9 residues of difference) with RmtD1 [77,79]. The *rmtD3* genes were found in two *P. aeruginosa* strains from Myanmar and Poland, respectively, and both of them were located on chromosomes [81,82]. Compared with RmtD3, there were 9 amino acid substitutions in RmtD and 4 amino acid substitutions in RmtD2 [81]. Recent studies have found that different gene subtypes have different transfer mechanisms. By the means of IS*26*-mediated recombinational events, *rmtD* with other genetic elements form complex transposons and may facilitate the future spread of the gene within *Enterobacteriaceae* [122]. *rmtD2* was located on transposon Tn*21*, and sequence analysis showed that the antimicrobial resistance region of *rmtD1* and *rmtD2* is formed through transposition or homologous recombination by IS*CR3* and IS*CR14* [79]. When it comes to *rmtD3* identified so far, it often resided in chromosomal mosaic regions, which comprise integrative and conjugative elements (ICEs) with variable cargo regions, carrying IS- or transposon-associated resistance genes [82]. Therefore, with the help of so many transfer elements, *rmtD* often occurred together with *bla*_SPM-1_, *bla*_KPC-2_, *bla*_TEM-1_, and *bla*_CTX-M-2_ [75,77,79]. We draw the Figure 2 to help you understand the genetic context of *rmtD* more comprehensively.

### 3.6. RmtE

The 16S-RMTase gene *rmtE* was first identified in cattle origin *E. coli* isolated from America in 2010 [84]. In 2014, a case of human infection by RmtE-producing *E. coli* was firstly reported in America and mediated resistance to aminoglycosides with MIC ≥ 256 μg/mL [85]. Obviously, the distribution of strains carrying *rmtE* was relatively simple and mainly exists in *E. coli*. The *rmtE2* was identified in swine-origin *E. coli* isolated from China, which existed in the IncI1 plasmid. More details about *rmtE2* were that a single base mutation of T→C is detected at nucleotide 20 of *rmtE*, which caused a replacement of Val (6) by Ala in the gene product [74]. *rmtE* was detected among *A. baumanii* from the UK. Compared with the sequences of *rmtE1* and *rmtE2* indicated that *rmtE3* had two SNPs: one at nucleotide 20 (T→C, Val 7 Ala) and another at nucleotide 141 (T→A, Asn 47 Lys) [87].

According to the genetic environment of *rmtE* and its alleles, there were several mobile genetic elements that probably mediated the transfer of *rmtE* between plasmids or between plasmids and chromosomes. *rmtE1* was identified on a *bla*_CMY-2_-carrying IncA/C plasmid called pYDC637 of *E. coli* in 2015. Within this unit, *rmtE1* was bound by an IS*CR20*-like element and an IS*1294*-like insertion sequence [123]. Interestingly, in 2017, *rmtE1* was identified on the chromosome of *E. coli* with a similar structure to the former. It had shown that the subunit containing *rmtE1* and its surrounding insertion sequences were similar to that of pYDC637 [86]. As to the genetic environment of *rmtE2*, an IS*CR20*-like transposase was located upstream of *rmtE2* and an IS*Vs1*-like transposase was located downstream [74]. As reported, there is an IS*Vs1*-like transposase located downstream of *rmtE3* [87]. In comparing the genetic context of *rmtE1* and *rmtE2*, IS*CR20*-like transposase was located upstream of *rmtE1* and *rmtE2*. However, the transposase genes located downstream of the 16S RMTase genes were distinct. IS*Vs1*-like transposase was located downstream of *rmtE2* and *rmtE3*. A general instance of the genetic environment surrounding *rmtE* is shown in Figure 2. To sum up, whether insertion sequences or broad-host-range, self-conjugative plasmids all played an important role in the initial mobilization of *rmtE* and the recombination with other resistance genes, such as *bla*_TEM-1_, *bla*_CMY-2_, *bla*_TEM-1_ [15,86,123].

### 3.7. RmtF

The 16S-RMTase gene *rmtF* was first identified in *K. pneumoniae* isolated from French in 2011, mediating resistance to aminoglycosides with MIC ≥ 256 μg/mL [89]. Subsequently, *rmtF* has been spread widely around the world, especially in *K. pneumoniae*, as well as can be found in *P. aeruginosa*, *E. coli,* and *C. freundii* [56,59,60,89]. In 2017, a new *rmtF* variant, *rmtF2*, was identified in *P. aeruginosa* in Nepal mediating resistance to aminoglycosides with MIC up to 1024 μg/mL [92]. Analysis of its predicted amino acid sequence reveals a substitution (Lys65Glu) compared with the sequence of RmtF [92].

Through whole-genome sequencing and analysis of antimicrobial resistance gene sequences, it was found that IS*CR5* was often located on both sides of *rmtF* and its allele *rmtF2* whether on plasmids or chromosomes, which used to be called insE as shown Figure 2 [89,92]; In recent years, transposase family genes and insertion sequence elements such as Tn*3*, Tn*1721*, IS*91*, and IS*6100* also have been found on both sides of *rmtF* or *rmtF2* [62,95]. *rmtF* is usually located on various conjugative plasmids, which belong to broad-host-range incompatibility groups such as IncA/C, IncR, IncFII, and IncFIB [58,90,94,95]. With the help of transferable plasmids and insertion sequences, *rmtF* was often associated with β-lactam and carbapenem resistance genes such as *bla*_NDM_, *bla*_CTX-M_, *bla*_OXA-232_, and *bla*_TEM-1_, especially in *K. pneumoniae* belongs to several high-risk clones ST231, ST147 [90,93,124,125]. The identification of RmtF coresident in strains harboring ESBLs, acquired AmpC enzymes, the NDM-type carbapenemases, and fluoroquinolone-resistance mechanisms not only leads to the potential for coselection and maintenance of resistance by the use of other antibiotics but also seriously compromises the treatment of life-threatening infections caused by Gram-negative organisms [56].

### 3.8. RmtG

The 16S-RMTase gene *rmtG* was first identified in *K. pneumoniae* isolated from Brazil in 2011, conferring resistance to aminoglycosides with MIC ≥ 256 μg/mL [80]. At present, the *rmtG* gene was still mainly prevalent in South America, especially in *K. pneumoniae* isolated from Brazil. It also could be found in *K. aerogenes, P. aeruginosa,* and *E. coli* isolated in America, India, Switzerland [3,59,96,97,98,100,101]. However, no strain carrying *rmtG* has been found in China so far.

Further analyses of the mobile genetic elements around the *rmtG* gene showed that *rmtG* was frequently located on the Tn*3* transposon of conjugative plasmids belonging to IncN, IncA/C types [80,96,102,126]. What is more, there was another rule that *rmtG* is part of an operon that includes genes related to rRNA and tRNA modification such as *rsmH*, *tgt*, and *rsmL* [101]. Around the multidrug resistance region, *rmtG* was flanked by IS*CR2* and IS*91*-like elements, which were responsible for the mobilization of the array [101,102]. Furthermore, the association of *rmtG* with IS*CR2* in a 2-fold tandem repeat suggested a gene amplification process [101]. An example of rmtG and its genetic context is shown in Figure 2. In addition, it was important to emphasize that the RmtG-producing strains such as *K. pneumoniae* were predominantly clonal complex 258 (CC258), which included sequence types such as ST11, ST258, ST437, and ST340 [35,80,96,97,127]. The coproduction of RmtG, ESBLs of the CTX-M type (eg., CTX-M-2, CTX-M-15, CTX-M-59) and carbapenemases (e.g., KPC-2) could further limit the treatment options for multidrug-resistant bacteria [35,96,101,127].

### 3.9. RmtH

The reports about *rmtH* were relatively bare. In 2013, the *rmtH* gene was first identified on a conjugative plasmid isolated from *K. pneumoniae* in Iraq, mediating high-level resistance to various aminoglycosides with MIC ≥ 256 μg/mL [103]. In 2017, a strain of *K. pneumoniae* carrying the *rmtH* gene isolated from Lebanon was reported and a complete sequence analysis was carried out. It was found that *rmtH* and *bla*_SHV-12_ were both located on the same IncFII plasmid, and integrated on the Tn*6329* transposon, with IS*26* and IS*CR2* insertion sequences on both sides involved in the formation of the multidrug resistance region [103,104]. You can find out the surrounding genetic elements of *rmtH* in Figure 2.

### 3.10. NpmA

The aminoglycoside-resistance 16S-RMTases were functionally divided into two subfamilies that modify the ribosome at either the N7 position of 16S rRNA nucleotide G1405 (m^7^G1405) or the N1 position of A1408 (m^1^A1408). Although enzymes from both subfamilies were found in aminoglycoside-producing bacteria, the m^7^G1405 methyltransferases were far more clinically prevalent than their m^1^A1408 methyltransferase counterparts [128].

NpmA was clinically isolated from *E. coli* strain ARS3 in Japan in 2007 [13]. Due to the posttranslational methylation of the A site of 16S rRNA at position A1408, NpmA leads to pan-aminoglycoside resistance encompassing both 4,5- and 4,6-disubstituted 2-DOS aminoglycosides with MIC ≥ 256 μg/mL, including neomycin and apramycin [12]. So, apramycin resistance seemed to be a good indicator for the detection of an A1408 16S-RMTase producer [13]. Later, the allele of *npmA* was identified in *C. difficile* and named *npmA2*. The study suggested that hospital-acquired *C. difficile* might be a reservoir for uncommon antibiotic resistance determinants such as *npmA* [105].

In 2021, NpmB was discovered in the process of screening for NpmA-like enzymes in the NCBI sequence databases, whose sequences were identified in *E. coli* genomes registered from the United Kingdom. NpmB1 and NpmB2 consisted of 217 amino acids, and only one amino acid substitution was identified in their sequences at position 21 (arginine for NpmB1 and cysteine for NpmB2). NpmB1 possesses 40% amino acid identity with NpmA1 and conferred resistance to all clinically relevant aminoglycosides, including 4,5-DOS agents. Phylogenetic analysis of NpmB1 and NpmB2, its single-amino-acid variant, revealed that the encoding gene was likely acquired by *E. coli* from a soil bacterium. The structure of NpmB1 suggested that it required a structural change of the β6/7 linker to bind to 16S rRNA. These findings established NpmB1 and NpmB2 as the second group of acquired pan-aminoglycoside resistance 16S-RMTases [12]. In the end, about the genetic context, an example of *npmA* is shown in Figure 2. However, we cannot find the sequence of *npmB*’s genetic context in the NCBI database, so the figure does not contain the *npmB* gene for the moment.

## 4. Conclusions

With the wide spread of multidrug-resistant bacteria and pan-drug resistant bacteria, monotherapy such as β-lactams or carbapenem for serious infections has become ineffective and unsuccessful. At this moment, the drug combination therapy containing kinds of antibiotics plays an irreplaceable role in the treatment of infectious diseases. As one of the first identified and used clinically, aminoglycoside antibiotics have a broad antimicrobial spectrum and can inhibit both Gram-negative bacteria and part of Gram-positive bacteria [5]. Aminoglycoside antibiotics regain the focus on the clinic as one of the effective options for the drug combination therapy [129].

However, a variety of resistance mechanisms lead to aminoglycosides resistance in bacteria. Among the mechanisms, the production of 16S-RMTases poses a potential threat. The reason for this statement is that they can confer not only complete resistance to a broad range of aminoglycosides but also high-level resistance that cannot be corrected by upregulating the dosage. In addition, several studies have shown that many subtypes of corresponding genotypes often lead to higher levels of resistance to aminoglycosides [81,92]; for example, *rmtF2* is able to mediate resistance to amikacin and arbekacin with MIC > 1024 μg/mL.

What is more, most of the 16S-RMTases encoding genes are often located on transferable plasmids which have a broad host range. With the development of whole-genome sequencing, it is abundantly clear that almost each 16S-RMTases gene is surrounded by some kinds of mobile genetic elements including transposons, integrons, and insertion sequences. Mobile genetic elements and conjugative plasmids not only promote resistance genes to transfer horizontally among all the members of *Enterobacteriaceae* but also upregulate the expression of the resistance genes resulting in higher MIC [102,120]. From another point of view, there is no doubt that mobile genetic elements also facilitate 16S-RMTases genes to associate with other resistance genes so that forming multidrug resistance regions. The coproduction of 16S-RMTases with ESBLs, AmpC enzymes, and carbapenemases largely limits the drug combination therapy for life-threatening infections caused by multidrug-resistant pathogens [121].

Up to now, there are 2 methods to screen the strain expressing 16S-RMTases: disk diffusion method shows that there is no inhibition zone around multiple aminoglycosides disk or the diameter of inhibition zone shrinking; microbroth dilution method shows that the high-level MIC of multiple aminoglycosides exceeds 128 μg/mL [14]. If the above results occur, it can be preliminarily determined that the strain carries 16S-RMTase genes. What is more, because of the difference in methylation site, acquired G1405 16S-RMTases and A1408 16S-RMTases confer resistance to different aminoglycosides [12]. For example, apramycin which belongs to 4,5-disubstituted 2-DOS can be used to screen A1408 16S-RMTase [13]. However, the above methods are still not routinely as well as widely applied to the clinical laboratory to detect the resistance phenotypes, enzyme types even corresponding subtypes of 16S-RMTase gene [121]. In the future, it is important to appeal to the clinician and researchers to place particular emphasis on the current situation of aminoglycosides resistance and explore optimized screening procedures and methods to detect the strains expressing 16S-RMTase as early as possible. Additionally, to control the prevalence and global dissemination of multidrug-resistant bacteria, it is highly recommended to evaluate the molecular epidemiology of aminoglycosides resistance genes and perform in-depth analysis of related mobile genetic elements.

## Figures and Tables

**Figure 1 antibiotics-11-00906-f001:**
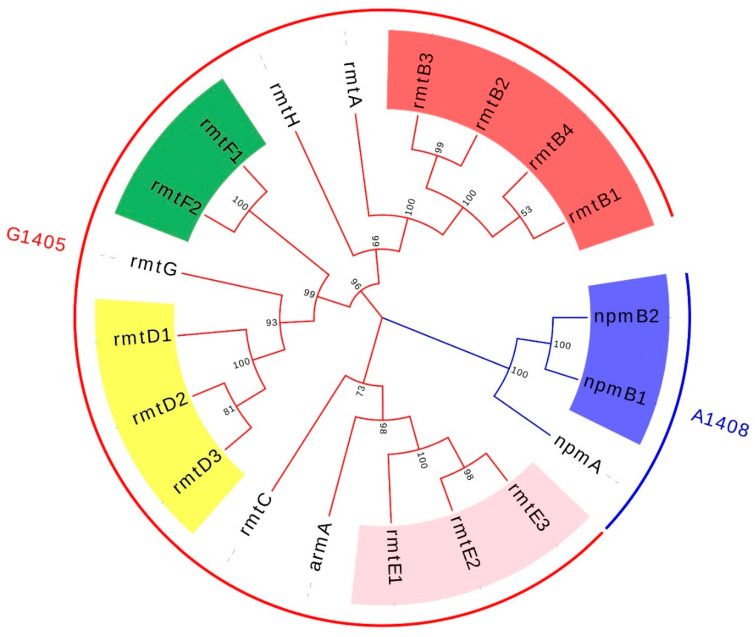
The phylogenetic reconstruction of 16S-RMTases encoding genes and their subtypes. The colors of the branch and external circle are typed by the action sites of 16S-RMTase. The subtypes of *rmtB*, *rmtD*, *rmtE*, *rmtF*, and *npmA* are highlighted with red, yellow, pink, green, and blue. GenBank: *rmtA*, NG_048057.1; *rmtB1*, NG_048058.1; *rmtB2*, NG_048059.1; *rmtB3*, NG_051535.1; *rmtB4*, NG_051536.1; *rmtC*, NG_048060.1; *rmtD1*, NG_048061.1; *rmtD2*, NG_050557.1; *rmtD3*, LC229698.1; *rmtE1*, NG_050558.1; *rmtE2*, NG_050559.1; *rmtE3*, MH572011.1; *rmtF1*, NG_048062.1; *rmtF2*, NG_051537.1; *rmtG*, NG_048064.1; *rmtH*, NG_048065.1; *npmA*, NG_048018.1; *npmB1*, NG_077965.1; *npmB2*, NG_077964.1.

**Figure 2 antibiotics-11-00906-f002:**
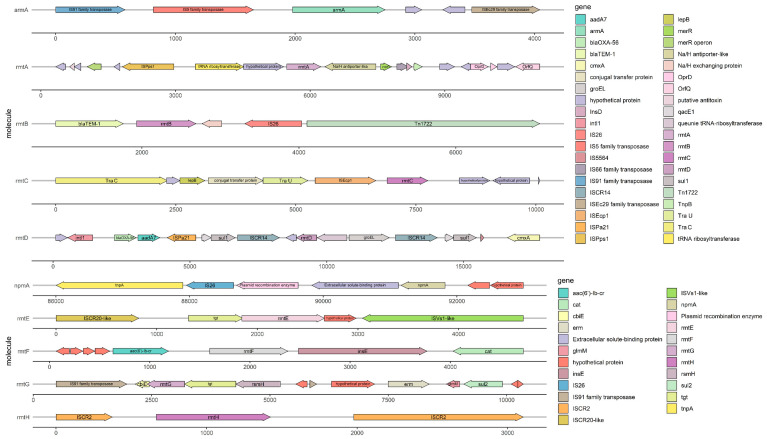
The sequence of the 16S rRNA methylase resistance gene and its genetic environment. GenBank: *rmtA*, AB083212.2; *rmtB*, FJ556899.1; *rmtC*, AB194779.2; *rmtD*, DQ914960.2; *rmtE*, KT428293.1; *rmtF*, JQ808129.1; *rmtG*, VLNW01000106.1; *rmtH*, KC544262.1; *armA*, AY220558.1; *npmA*, AB261016.2.

**Table 1 antibiotics-11-00906-t001:** The strains with 16S rRNA methylase resistance gene and their distribution.

Genotype	Action Site	MIC (μg/mL)	Subtype	Epidemic Strains	Year of Isolation and Distribution	References
*armA*	G1405	AMK ≥ 256 GEN ≥ 256 TOB ≥ 256 KAN ≥ 256		*Klebsiella pneumoniae*	2019-Japan, Brazil, Greece, India 2020-China; 2021-Britain; 2022-China	[7,15,16,17,18,19,20,21,22,23,24,25,26,27]
				*Escherichia coli*	2020-India, French; 2021-Britain	
				*Acinetobacter baumanii*	2018-China; 2020-Greece, Brazil; 2021-Britain	
				*Enterobacter cloacae*	2020-Myanmar; 2021-Britain	
				*Citrobacter freundii*	2021-Britain	
				*Salmonella enterica*	2021-China	
				*Serratia marcescens*	2020-Turkey	
				*Enterobacter xiangfangensis*	2020-Myanmar	
*rmtA*	G1405	ABK ≥ 512 AMK ≥ 512 GEN ≥ 512 KAN ≥ 512 TOB ≥ 512		*Pseudomonas aeruginosa*	2003-Japan; 2009-Korea; 2014-Japan; 2017-China	[9,28,29,30,31,32,33,34]
				*K. pneumoniae*	2011-Switzerland; 2021-China	
				*E. coli*	2019-Iran; 2020-India	
				*E. cloacae*	2017-China	
*rmtB*	G1405	AMK > 256 GEN > 256 KAN > 256 TOB > 256	*rmtB1*	*E. coli*	2019-Japan, China, Korea; 2020-India, China, French; 2021-China, Britain, Greece, Kenya; 2022-Switzerland	[7,20,24,32,35,36,37,38,39,40,41,42,43,44,45,46,47,48,49,50,51,52,53,54,55]
				*K. pneumoniae*	2019-China, Japan; 2020-China, Japan; 2021-Britain, Brazil, China; 2022-Switzerland	
				*S. enterica*	2020-China; 2021-China; 2022-China	
				*E. cloacae*	2019-Iran; 2020-China	
				*A. baumanii*	2020-Brazil	
				*C. freundii*	2021-China	
				*Providencia stuartii*	2021-Britain, Greece	
				*Proteus mirabilis*	2021-Greece, China	
				*Klebsiella variicola*	2021-China	
			*rmtB2*	*Providencia rettgeri*	2021-America	
			*rmtB4*	*P. aeruginosa*	2019-Myanmar; 2021-Egypt, Britain, India	
*rmtC*	G1405	AMK > 256 GEN > 256 TOB ≥ 512 ABK = 1024		*P. mirabilis*	2006-Japan; 2008-Australia	[15,35,56,57,58,59,60,61,62,63,64,65,66,67,68,69,70,71,72,73]
				*K. pneumoniae*	2013-India, Nepal; 2017-Vietnam 2019-Iran, Albania; 2020-Turkey	
				*E. coli*	2013-India; 2016-Turkey; 2020-India; 2021-Britain 2022-Switzerland	
				*E. cloacae*	2014-South Africa; 2019-Saudi Arabia; 2021-Britain 2022-Switzerland	
				*S. enterica*	2010-Britain; 2017-China	
				*Salmonella stanley*	2013-China	
				*C. freundii*	2014-South Africa; 2021-Britain; 2022-Switzerland	
				*Morganella morganii*	2022-Switzerland	
				*P. aeruginosa*	2015-India; 2022-Britain	
				*S. marcescens*	2014-South Africa	
				*A. baumanii*	2018-Ukraine; 2020-Brazil	
				*Klebsiella aerogenes*	2019-China	
				*E. xiangfangensis*	2019-Myanmar; 2022-Switzerland	
*rmtD*	G1405	AMK > 256 ABK > 256 GEN > 256 TOB > 256	*rmtD1*	*K. pneumoniae*	2008-Argentina, Chile, Brazil; 2013-America 2020-Turkey; 2021-Brazil	[55,61,62,74,75,76,77,78,79,80,81,82,83]
				*E. coli*	2020-India	
				*E. cloacae*	2008-Chile; 2010-Argentina	
		AMK ≥ 256 KAN ≥ 256 GEN ≥ 1024 GEN ≥ 1024	*rmtD2*	*C. freundii*	2010-Argentina	
				*Enterobacter aerogenes*	2010-Argentina	
		ABK > 1024 AMK > 1024 GEN > 1024 KAN > 1024 TOB > 1024	*rmtD3*	*P. aeruginosa*	2007-Brazil; 2011-Brazil; 2018-Myanmar 2019-Poland; 2021-Brazil; 2022-Britain	
*rmtE*	G1405	AMK ≥ 256 GEN ≥ 256 KAN > 256 TOB ≥ 256	*rmtE1*	*E. coli*	2010-America; 2014-America; 2016-China; 2017-America; 2020-India	[55,74,84,85,86,87,88]
				*P. aeruginosa*	2019-Myanmar	
				*E. xiangfangensis*	2022-Myanmar	
			*rmtE2*	*E. coli*	2015-China	
			*rmtE3*	*A. baumanii*	2022-Britain	
*rmtF*	G1405	AMK ≥ 256 GEN ≥ 256 TOB ≥ 256 APR = 2	*rmtF1*	*K. pneumoniae*	2012-French; 2013-India, Nepal, Britain; 2014-America; 2015-India; 2016-Egypt; 2018-Switzerland, Britain, Ireland; 2020-China; 2020-India; 2022-Switzerland	[35,36,56,57,58,59,60,62,89,90,91,92,93,94,95]
				*E. coli*	2013-India; 2015-India; 2020-India	
				*P. aeruginosa*	2015-India; 2021-Egypt	
				*C. freundii*	2013-India; 2015-India	
				*E. cloacae*	2013-India; 2014-South Africa	
				*Citrobacter kooseri*	2015-India	
				*P. mirabilis*	2015-India	
		AMK > 1024 ABK > 1024	*rmtF2*	*P. aeruginosa*	2017-Nepal; 2019-Myanmar; 2022-Britain	
*rmtG*	G1405	AMK > 256 TOB > 256 GEN > 256 ABK > 256		*K. pneumoniae*	2013-Brazil; 2014-America; 2014-Chile 2015-India; 2016-Brazil; 2017-Switzerland 2022-Switzerland	[35,59,80,96,97,98,99,100,101,102]
				*P. aeruginosa*	2015-Brazil	
				*E. coli*	2015-India; 2020-India	
				*K. aerogenes*	2019-Brazil	
				*Enterobacter hormaechei*	2020-Brazil	
*rmtH*	G1405	GEN > 256 TOB > 256 AMK > 256 ABK > 256		*K. pneumoniae*	2013-Iraq; 2017-Lebanon	[103,104]
				*E. coli*	2020-India	
*npmA*	A1408	KAN > 256 TOB > 256 NEO > 256 APR > 256	*npmA1*	*E. coli*	2007-China, Japan	[13,31,105,106]
				*P. aeruginosa*	2018-Japan	
				*K. pneumoniae*	2018-Japan	
			*npmA2*	*Clostridioides difficile*	2019-America	
*npmB*	A1408	AMK = 64 TOB = 128 GEN = 32 NEO = 128 APR > 256	*npmB1* *npmB2*	*E. coli*	2021-Britain	[12]

GEN, Gentamicin; TOB, Tobramycin; AMK, Amikacin; ABK, Arbekacin; NEO, Neomycin; APR, Apramycin; KAN, Kanamycin. Due to space limitations, *armA* and *rmtB* genes only list the distribution of epidemic strains around the world from 2019 to 2022, and there is no strain carrying *rmtB3* found from 2019 to 2022.

## Data Availability

Publicly available datasets were analyzed in this study. This data can be found here: https://www.ncbi.nlm.nih.gov/ (accessed on 17 June 2022).

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
