# Peer review of "Research Updates of Plasmid-Mediated Aminoglycoside Resistance 16S rRNA Methyltransferase"

_antibiotics, 2022, doi:10.3390/antibiotics11070906_

Round 1
Reviewer 1 Report
The manuscript "Research updates of plasmid-mediated aminoglycoside resistance genes" by W. Yang and F. Hu is a review of 16S rRNA methyltransferase-associated aminoglycoside resistance. The work sounds original and actual due to the antimicrbial resistance becomes important medical, social, scientific and even political issue. Guess, the work could be accepted for publication after moderate revision. Some points are listed below:
1. The title and abstract should be carefully checked and partially rewrited. First, the main mechanism of aminoglycoside resistance covered by the review is the methylation of drug target (16S rRNA) by specific enzymes. It should be highlighted in the title and abstract.
"plasmid-mediated aminoglycoside resistance" is not clear and representative due to we know a lot of plasmid and transposon-mediated aminoglycoside-modifying enzymes.
line 8: ...widespred...widely... - please, rewrite without repeats;
lines 9-10: But in recent years, the emergence of aminoglycosides resistance genes... - The aminoglycoside resistance is the old problem, it's strange to see this. The modern and discussed problem is the emergence of pan-aminoglycoside resistance (or high-level resistance to aminoglycosides).
lines13-16: In this review, we will make a presentation on the prevalence and genetic environment of aminoglycoside resistance genes which lead to high-level resistance to aminoglycosides, especially 16S rRNA methyltransferase encoding genes. - Better will be "of 16S rRNA methyltransferase encoding genes that lead to high-level resistance to aminoglycosides". Your review focused on the 16S rRNA methyltransferase-associated resistance, not "especially" but fully.
2. Minor revisions:
line 55: NH2 - "2" should be subscript;
line 317: MIC.[108,126] - the reference should be before the dot;
line 322: pathogens.[127] - the reference should be before the dot;
line 329: 128μg/ml.[136] - the reference should be before the dot;
line 334: transferase.[13] - the reference should be before the dot;
3. The figure 2 is important but it's absolutely non-readable. Please, format the figure or make it bigger and transfer to the supplementary.
Author Response
Thank you for taking time out of your busy schedule to review the manuscript "Research updates of plasmid-mediated aminoglycoside resistance genes". Now we have carefully corrected and replied to the manuscript for this revision.
Response to Reviewer 1
Point 1: The title and abstract should be carefully checked and partially rewrited. First, the main mechanism of aminoglycoside resistance covered by the review is the methylation of drug target (16S rRNA) by specific enzymes. It should be highlighted in the title and abstract.
Response 1: Thanks for the brilliant comments. The key point, 16S rRNA methyltransferase,
are indeedly not displayed very well in the title and abstract. So in this revision, we rewrite the title as “Research updates of aminoglycoside resistance due to plasmid-mediated 16S rRNA methyltransferase”. And about the abstract, we emphasize “the production of 16S rRNA methyltransferase ” as the main mechanism of aminoglycosides resistance covered in this review.
Point 2: "plasmid-mediated aminoglycoside resistance" is not clear and representative due to we know a lot of plasmid and transposon-mediated aminoglycoside-modifying enzymes.
Response 2: Thanks for your suggestions. It is true that “plasmid-mediated aminoglycoside resistance” is confusing in title, cause another important aminoglycosides resistance mechanism, aminoglycosides-modifying enzymes, are commonly mediated by a lot of plasmid and transposon too. So, on account of your proposal, we refine the title by adding “16S rRNA methyltransferase” and stress the role of plasmid in the transfer of 16S rRNA methyltransferase encoding genes in the later section of the review.
Point 3: line 8: ...widespred...widely... - please, rewrite without repeats;
Response 3: Thanks for your meticulous reviewing. I’m sorry for my unskilled English writing, and I have replaced the “widely” by “a variety of” to express the meaning more smoothly.
Point 4: lines 9-10: But in recent years, the emergence of aminoglycosides resistance genes... - The aminoglycoside resistance is the old problem, it's strange to see this. The modern and discussed problem is the emergence of pan-aminoglycoside resistance (or high-level resistance to aminoglycosides).
Response 4: Thanks for your professional suggesitons. It is true that aminoglycoside resistance is an old topic. But in recent years, because of the emergence of 16S rRNA methyltransferase, high-level resistance to pan-aminoglycosides becomes prevalent and intractable. So I do agree with your opinion and reorganise my words in the abstract.
Point 5: lines13-16: In this review, we will make a presentation on the prevalence and genetic environment of aminoglycoside resistance genes which lead to high-level resistance to aminoglycosides, especially 16S rRNA methyltransferase encoding genes. - Better will be "of 16S rRNA methyltransferase encoding genes that lead to high-level resistance to aminoglycosides". Your review focused on the 16S rRNA methyltransferase-associated resistance, not "especially" but fully.
Response 5: Thanks for you proposals. Your suggestion do remind me that the key point of the review is “16S rRNA methyltransferase” and we should put more emphasis on this point. So I rewrite the last sentence of abstract shown that “In this review, we will make a presentation on the prevalence and genetic environment of 16S rRNA methyltransferase encoding genes which lead to high-level resistance to aminoglycosides”.
Point 6: Minor revisions:
line 55: NH2 - "2" should be subscript;
line 317: MIC.[108,126] - the reference should be before the dot;
line 322: pathogens.[127] - the reference should be before the dot;
line 329: 128μg/ml.[136] - the reference should be before the dot;
line 334: transferase.[13] - the reference should be before the dot;
Response 6: Thanks for you attentive reviewing. I’m very sorry for my incorrect format of referrence, the above errors and similar problem in this review have been carefully corrected.
Point 7: The figure 2 is important but it's absolutely non-readable. Please, format the figure or make it bigger and transfer to the supplementary.
Response 7: Thanks for your affirmation of the figure 2. I format the figure again and improve the resolusion of figure 2 and transfer to the supplementary.
Reviewer 2 Report
The review article covered research on aminoglycoside resistance mechanisms and resistance genes. The authors started with a brief history of the antibiotic and its use in the clinics, followed by classification and resistance mechanisms and detailed discussions on a panel of resistance genes. Finally, the authors concluded the review with future research directions to detect and treat antibiotic-resistant bacterial infections.
Overall, the review covered an interesting and important topic. The flow of the review article is straightforward. However, here are a few major and minor issues.
Major issues:
- The title emphasized ‘plasmid-mediated’ aminoglycoside resistance genes, but the bulk of the review and the abstract lack a precise focus on ‘plasmid-mediated.’ I suggest including its definition and general mechanisms in Part 3. It will also be helpful to discuss if ‘plasmid-mediated’ antibiotic resistance is more prevalent in aminoglycosides than in other classes of antibiotics.
- The authors did not provide a description and explanation of the two figures. For example, Figure 1 is mentioned in lines 79-81 as ‘the phylogenetic reconstruction of 16S rRNA methyltransferases and their subtypes is shown in Figure 1’. The authors need to provide more information, such as the implications of the phylogenetic tree. Figure 2 can be referred to in the discussion on each gene.
- There is little connection between paragraphs and sentences. Consequently, it is sometimes difficult to follow the authors’ arguments. For example, in lines 323-330, the authors started with ‘no routine or method …to detect resistance in the clinical labs’, but two methods were immediately described. It is unclear if the authors want to emphasize that laboratory techniques are not widely applicable in the clinics or some other ideas without a clear connective sentence or phrase.
Minor issues:
- The writing is wordy and confusing in many places. For example, I have trouble understanding the statement in lines 33-35, 307-309, the term ‘the genetic environment structures’ in lines 90-91, and ‘consistent resistance’ in line 61.
- Some sentences contain grammar errors. For example, ‘French’ in line 127 and ‘usually often’ in line 141.
Author Response
Thank you for taking time out of your busy schedule to review the manuscript "Research updates of plasmid-mediated aminoglycoside resistance genes". Now we have carefully corrected and replied to the manuscript for this revision.
Response to Reviewer 2
Point 1: The title emphasized ‘plasmid-mediated’ aminoglycoside resistance genes, but the bulk of the review and the abstract lack a precise focus on ‘plasmid-mediated.’ I suggest including its definition and general mechanisms in Part 3. It will also be helpful to discuss if ‘plasmid-mediated’ antibiotic resistance is more prevalent in aminoglycosides than in other classes of antibiotics.
Response 1: Thanks for you brilliant suggestions. It is true that the description of “plasmid-mediated” in the bulk of the review is deficient and not precise. According to your proposal, we put a special description of the definition and general mechanisms about “plasmid-mediated” in Part 3. It is shown that “Except for the high-level antimicrobial resistance, the 16S rRNA methyltransferase encoding genes found so far are mostly located within transferable plasmid and/or associated with mobile genetic elements such as transposons, integrons, and insertion sequence. Furthermore, with the support of plasmid and mobile genetic elements, 16S rRNA methyltransferase encoding genes are often associsted with other resistance genes, lead-ing multi-drug resistance even pan-drug resistance.”
Point 2: The authors did not provide a description and explanation of the two figures. For example, Figure 1 is mentioned in lines 79-81 as ‘the phylogenetic reconstruction of 16S rRNA methyltransferases and their subtypes is shown in Figure 1’. The authors need to provide more information, such as the implications of the phylogenetic tree. Figure 2 can be referred to in the discussion on each gene.
Response 2: Thanks for your reminder. It is true that “the phylogenetic reconstruction of 16S rRNA methyltransferases and their subtypes is shown in Figure 1 ” is a little bit abrupt. So we put more description about the Figure 1 shown that “In order to figure out the evolutionary relationships of 16S rRNA methyltransferases en-coding genes and their subtypes, a phylogenetic tree is reconstructed and shown in Figure 1.” And we put the reference of Figure 2 in the discussion on each gene, such as line166-167 “In Figure2, you can acquire more details of the rmtA and its genetic context which is shown as an example.”
Point 3: There is little connection between paragraphs and sentences. Consequently, it is sometimes difficult to follow the authors’ arguments. For example, in lines 323-330, the authors started with ‘no routine or method …to detect resistance in the clinical labs’, but two methods were immediately described. It is unclear if the authors want to emphasize that laboratory techniques are not widely applicable in the clinics or some other ideas without a clear connective sentence or phrase.
Response 3: Thanks for your suggestions. I’m sorry for my ambiguous expression. It is indeed that we want to emphasize that the mentioned two laboratory methods are still not widely applicable in the clinics. So we reorganise the words in Part 4 that “However, the above methods are still not routinely as well as widely applied to the clinical laboratory to detect the resistance phenotypes, enzyme types even corresponding subtypes of 16S rRNA methyltransferase gene. In the future, it is very important to appeal to the clinician and researchers to place particular emphasis on the current situation of aminoglycosides resistance and explore optimized screening procedures and methods to detect the strains expressing 16S rRNA methyltransferase as early as possible.”
Point 4: The writing is wordy and confusing in many places. For example, I have trouble understanding the statement in lines 33-35, 307-309, the term ‘the genetic environment structures’ in lines 90-91, and ‘consistent resistance’ in line 61.
Response 4: Thanks for your reminder. I’m sorry for my wordy and confusing writing. Please, give me the chance to explain the meaning of the sentence in lines 33-35. Actually, we want to express that in the past, because of the by-side effect of aminoglycosides, the usage of this kind of antibiotics had been largely limited. However, with the emergence of multi-drug resistance bacteria, a drug combination therapy containing aminoglycosides can ideally take better advantage of this class of anitibiotic. In the meantime, because of the combination with other antibiotics, the dosage of aminoglycosides can be adjusted to decreasing the toxicity. About the lines 307-309, what we are trying to say that the subtypes of 16S rRNA methyltransferase encoding genes usually mediate higher resistance to aminoglycosides, with MIC>1024μg/mL. So, in order to express more clearly, we rewrite the sentence respectively, and show that in lines 34-38 “Additionally, aminoglycosides can be used as one of the effective antimicrobial combi-nations for the treatment of life-threatening infections caused by multidrug-resistant bacteria. Because aminoglycosides can be used in combination with other antibiotics, the toxicity can be decreased by adjusting the dosage.” And in lines 372-375 “In addition, several studies have shown that many subtypes of corresponding genotypes often lead to higher levels of resistance to aminoglycosides; for example, rmtF2 is able to mediate resistance to amikacin and arbekacin with MIC>1024μg/mL”. The term “the genetic environment structures” and “consistent resistance” we also replace with “the genetic context” in line 133 and “high-level resistance against pan-aminoglycoside” in lines 81-82.
Point 5: Some sentences contain grammar errors. For example, ‘French’ in line 127 and ‘usually often’ in line 141.
Response 5: Thanks for your correction. I’m sorry for my incorrect writing, expression and grammatical errors in this review have been carefully corrected.
Reviewer 3 Report
1. The English need improvement since there are some grammatical and syntax errors in the manuscript. For example,
· in line number 81, the words “ubtypes is” may be as “ubtypes are”;
· in line number 130, “often” as “was often”;
· in line number 152, “located” as “is located”;
· in line number 197, “probably” as “that probably”;
· in line number 204 and 205, “located” as “is located”;
· in line number 255, “usually” as “is usually”;
· in line number 261, “both” as “are both”.
The grammar mistakes which are not mentioned here are also to be checked and corrected properly.
2. There are some typing mistakes as well, and authors are advised to carefully proof-read the text. For example,
· in line number 80, the word “reconstruction” may be as “reconstructions”;
· in line number 118, “researches” as “research”;
· in line number 182, “occured” as “occurred”;
· in line number 190, “existeded” as “existed”.
The typos not mentioned here are also to be checked and corrected properly.
3. Check the abbreviations throughout the manuscript and introduce the abbreviation when the full word appears the first time in the text as well as abstract and then use only the abbreviation (For example, deoxystreptamine (DOS), MIC, etc.,). Make a word abbreviated in the article that is repeated at least three times in the text, not all words need to be abbreviated.
4. The references are not arranged properly in a uniform format and they should be carefully checked and corrected as per the journal instructions. For example, reference number 45 and 61 the journal names are given in capital letters and it should be corrected properly.
5. In the conclusion seems to be in general, it is highly recommended to include limitations of the study and potential future research goals.
Author Response
Thank you for taking time out of your busy schedule to review the manuscript "Research updates of plasmid-mediated aminoglycoside resistance genes". Now we have carefully corrected and replied to the manuscript for this revision.
Response to Reviewer 3
Point 1: The English need improvement since there are some grammatical and syntax errors in the manuscript. For example,
in line number 81, the words “ubtypes is” may be as “ubtypes are”;
in line number 130, “often” as “was often”;
in line number 152, “located” as “is located”;
in line number 197, “probably” as “that probably”;
in line number 204 and 205, “located” as “is located”;
in line number 255, “usually” as “is usually”;
in line number 261, “both” as “are both”.
The grammar mistakes which are not mentioned here are also to be checked and corrected properly.
Response 1: Thanks for your attentive reviewing. I’m sorry for my incorrect writing, the grammar errors and similar mistakes in this review have been checked and corrected.
Point 2: There are some typing mistakes as well, and authors are advised to carefully proof-read the text. For example,
in line number 80, the word “reconstruction” may be as “reconstructions”;
in line number 118, “researches” as “research”;
in line number 182, “occured” as “occurred”;
in line number 190, “existeded” as “existed”.
The typos not mentioned here are also to be checked and corrected properly.
Response 2: Thanks for your suggestions. I’m sorry for my carelessness and unskilled English writing. I do agree with your proposal and carefully correct one by one.
Point 3: Check the abbreviations throughout the manuscript and introduce the abbreviation when the full word appears the first time in the text as well as abstract and then use only the abbreviation (For example, deoxystreptamine (DOS), MIC, etc.,). Make a word abbreviated in the article that is repeated at least three times in the text, not all words need to be abbreviated.
Response 3: Thanks for your recommendation. It is true that the abbreviations of this review are non-standard. We check the abbreviations throughout the manuscript again and introduce the abbreviation the definition “deoxystreptamine (DOS)”, “minimal inhibitory concentration (MIC) ” and “16S rRNA methyltransferase (16S-RMTase)”.
Point 4: The references are not arranged properly in a uniform format and they should be carefully checked and corrected as per the journal instructions. For example, reference number 45 and 61 the journal names are given in capital letters and it should be corrected properly.
Response 4: Thanks for your reminder. According to your suggestions, we carefully check the referrences and arrange them in a uniform format.
Point 5: In the conclusion seems to be in general, it is highly recommended to include limitations of the study and potential future research goals.
Response 5: Thanks for your suggestions. It is true that the conclusion should include the limitations of the study and potential future research goals. So in the later section of the conclusion, we add this part in lines 412-421 as “However, the above methods are still not routinely as well as widely applied to the clin-ical laboratory to detect the resistance phenotypes, enzyme types even corresponding subtypes of 16S-RMTase gene. In the future, it is very important to appeal to the clinician and researchers to place particular emphasis on the current situation of amino-glycosides resistance and explore optimized screening procedures and methods to de-tect the strains expressing 16S-RMTase as early as possible. Additionally, to control the prevalence and global dissemination of multidrug-resistant bacteria, it is highly recom-mended to evaluate the molecular epidemiology of aminoglycosides resistance genes and perform in-depth analysis of related mobile genetic elements.”